# Evaluation of the Capsaicinoid Extraction Conditions from Mexican *Capsicum chinense* Var. Mayapan with Supercritical Fluid Extraction (SFE)

Kevin Alejandro Avilés-Betanzos [1] , Matteo Scampicchio [2] , Giovanna Ferrentino [2] ,
Manuel Octavio Ramírez-Sucre [1] and Ingrid Mayanin Rodríguez-Buenfil [1,*]

[1] Centro de Investigación y Asistencia en Tecnología y Diseño del Estado de Jalisco A.C., Subsede Sureste, Tablaje Catastral 31264, Km. 5.5 Carretera Sierra Papacal-Chuburná Puerto, Parque Científico Tecnológico de Yucatán, Mérida 97302, Mexico; keaviles_al@ciatej.edu.mx (K.A.A.-B.); oramirez@ciatej.mx (M.O.R.-S.)

[2] Faculty of Science and Technology, Free University of Bolzen-Bolzano, Piazza Università 1, 39100 Bolzano, Italy; matteo.scampicchio@unibz.it (M.S.); giovanna.ferrentino@unibz.it (G.F.)

[*] Correspondence: irodriguez@ciatej.mx

**Abstract:** Capsaicin (Cp) is a secondary metabolite produced by the *Capsicum* plant family. This molecule exhibits various biological properties such as antioxidant capacities, anti-obesogenic effects, and antidiabetic properties, among others. However, conventional extraction methods for Cp present several disadvantages including toxicity, extraction time, and low purity. Therefore, the utilization of supercritical fluid extraction techniques represents a viable option for obtaining highly pure and low-toxicity oleoresins (capsaicin-rich extracts). This approach involves the use of $CO_2$ in the supercritical state and finds applicability in the pharmaceutical, food, and cosmetic industries. The *Capsicum chinense* variety from the Yucatán Peninsula is a crop with significant economic impact in the region, due to having the highest concentrations of Cp in Mexico. This significant characteristic is attributed to its adaptation to the unique conditions (climate, soil, solar radiation, humidity) of the southeastern region of Mexico. The objective of this study was to evaluate the effect of temperature (45 °C, 60 °C), pressure (1450 psi, 2900 psi), and extraction time (60 min, 120 min) on the supercritical fluid extraction of Cp and dihydrocapsaicin (DhCp) from *Capsicum chinense* Jacq. The results obtained demonstrated that the extraction conditions of 45 °C, 1450 psi, and 60 min yielded the highest concentration of Cp ($37.09 \pm 0.84$ mg/g extract) and DhCp ($10.17 \pm 0.18$ mg/g extract), while the highest antioxidant capacity ($91.48 \pm 0.24$% inhibition) was obtained with 60 °C, 2900 psi, and 60 min. The findings of this study indicate that the lower the pressure and extraction time, the higher the concentrations of Cp and DhCp compared to previous reports. This represents an opportunity for cost reduction in production lines and improved utilization of *Capsicum chinense* in the agrifood industry through additional optimization processes.

**Keywords:** *Capsicum chinense*; supercritical fluid extraction; capsaicinoids; antioxidant capacity

## 1. Introduction

In Mexico, commercialization of fruits from the *Capsicum* genus (*Solanaceae* family) of plants have become relevant due to its cultural, culinary (sauces, powder), and commercial value [1]. The Mexican export of chili peppers has experienced a notable increase in recent years, resulting in imports of only 30% of the production. This achievement has positioned Mexico as the third-highest seller in the international market, with the Mexican chili pepper market reaching a value of $984.7 million USD [2,3].

The habanero pepper, or *Capsicum chinense* Jacq., is one of the primary chilies produced and exported by Mexico, with the highest production located in the Yucatán Peninsula. This geographical area encompasses the states of Campeche, Quintana Roo, and Yucatán and has unique climatic, edaphological, anthropological, and other conditions unparalleled in

the world [1]. These exceptional conditions allow the habanero pepper to develop unique characteristics on a global scale. In 2010, the habanero pepper from the Yucatán Peninsula was granted the designation of origin due to its unparalleled attributes [4], and since that year, various industries have shown interest in the habanero pepper produced in the Yucatán Peninsula, leading to the development of different techniques for the extraction and analysis of its secondary metabolites [5]. Cp is the most significant secondary metabolite for various industries such as food, pharmaceuticals, cosmetics, and agriculture, among others. This alkaloid is of great interest due to its bioactive characteristics, with pungency (spiciness) being the most extensively studied [6,7]. Indeed, Cp has been reported to have antioxidant and radioprotective capabilities [8], anti-obesogenic effects [9], as well as antimicrobial, antifungal, and antiviral properties [5]. Additionally, Cp has demonstrated potential for neuropathic pain control and gastroprotective effects [10], among others.

To obtain capsaicin-rich extracts, also known as oleoresins, various extraction techniques can be implemented, such as (1) maceration, with different organic solvents such as ethanol, acetone, or acetonitrile in a biomass:solvent loading ratio of 15 g:100 mL—for 24 h [11]; (2) Soxhlet extraction, with methanol at a ratio of 1:50 *w/v* (*Capsicum annum*:Solvent) and an extraction time of 2 h; (3) ultrasound-assisted extraction (25 mL methanol per 1 g of sample, 10 min); (4) pressurized liquid extraction (methanol, 100 °C, 1500 psi); (5) microwave-assisted extraction, by using 25 mL of ethanol (solvent) per 0.5 g of sample, with 500 W of power for five minutes [12]; (6) using a tunable aqueous polymer-phase impregnated resin (TAPPIR), by preparing an aqueous two-phase system and sonicating for 60 min and finally adding 50 mL of ethyl acetate to extract the capsaicinoids in a constant shake for 30 min [13]; (7) using enzymatic treatment, with a *R. nigricans* fungus using 100 mL of potato dextrose agar (PDA) at 28 °C for 24 h [14]; or (8) supercritical fluid extraction, using $CO_2$ as a solvent and extraction conditions of 15 Mpa and 40 °C for 5 h [15]. Supercritical fluid extraction (SFE) is a green technology, generally recognized as safe (GRAS) that has been demonstrated as an efficient extraction technique to extract oleoresins with a high concentration of Cp from fruits of the *Capsicum* family with high quality and purity. Additionally, this technique avoids the use of highly toxic organic solvents (such as methanol, acetonitrile, petroleum ether, etc.) by substituting them with $CO_2$, a highly pure, non-flammable, easy-to-handle, low-cost, and non-toxic solvent that is easily removed from extracts due to its chemical nature and to the extraction process. This provides an advantage when compared to other extraction techniques when used in the food, pharmaceutical, cosmetic, and other industries [16–18].

SFE has been used to extract high amounts of Cp from other chili pepper matrices such as the red pepper [19], biquinho pepper [20] and malagueta pepper [15,21]. On the other hand, regarding habanero pepper, Rocha-Uribe et al. [22] only reported the operational cost for Cp extraction from habanero peppers from the Yucatán Peninsula, using the extraction conditions of 1450.37 psi and 35 °C for 60 min (extraction time). Currently, there is no information available regarding the evaluation of Cp extraction conditions via SFE from habanero peppers from the Yucatán Peninsula.

The objective of this study was to evaluate the extraction conditions such as temperature, pressure, and extraction time using supercritical fluids for extracting capsaicinoids from habanero peppers from de Yucatán Peninsula.

## 2. Materials and Methods

### 2.1. Raw Materials

Habanero peppers (*Capsicum chinense* Jacq. Var. Mayapan) were grown in a controlled greenhouse environment, in the Chicxulub Pueblo town, located in Yucatán, Mexico. The geographical coordinates were 21°08′50.5″ N and 89°29′42.8″ W. The cultivation occurred in lithic leptosol soil (World Reference Base for soil resources classification). The habanero pepper fruits were harvested in December 2019. The peppers were harvested in an immature state (green color).

### 2.2. Habanero Pepper Drying and Sieved Process

The habanero peppers used in the present study were those at the immature state (green color); for this reason, the classification of the habanero pepper required the separation of immature habanero pepper (green color) fruits from the mature ones (orange color), and from those with physical damage as well as any agronomic waste (leaves, stems, and peduncles).

The selected habanero peppers (immature state) were subjected to a drying process at a temperature of 65 °C for 72 h, utilizing a FELISA oven (model FE-292) [23]. After drying, the habanero peppers were ground using a blender (Oster®, Mexico City, Mexico). The resulting powder was sieved using a #35 (500 µm particle size) sieve. The habanero pepper powder was stored in aluminum foil-lined plastic bags at room temperature until use.

### 2.3. Habanero Pepper Capsaisinoids Extraction

### 2.3.1. Experimental Design

A $2^3$ factorial design was employed to evaluate the effect of temperature ($X_1$), pressure ($X_2$), and extraction time ($X_3$) on the extraction process of capsaicinoids in the habanero pepper. For each factor, two levels were taken into consideration. For temperature (Tp), the lower level was set to 45 °C ($-1$), while the higher level was established at 60 °C (1). Regarding pressure (Ps), the experimental conditions consisted of 1450 psi ($-1$) as the low level and 2900 psi (1) as the high level. As for the extraction time (Et), 60 min ($-1$) and 120 min (1) were established as the low and high levels, respectively (Table 1).

**Table 1.** Real and encoded values of the factors used in the $2^3$ experimental design for the evaluation of capsaicinoid extraction conditions from habanero pepper (*Capsicum chinense* Jacq.) with SFE.

| Factors | Encoded Factors | Encoded Values | | Real Values | |
|---|---|---|---|---|---|
| | | Low Level | High Level | Low Level | High Level |
| Temperature (°C) | $X_1$ | $-1$ | 1 | 45 | 60 |
| Pressure (psi) | $X_2$ | $-1$ | 1 | 1450 | 2900 |
| Extraction time (min) | $X_3$ | $-1$ | 1 | 60 | 120 |

The response variables measured in this study were Cp, DhCp, and total capsaicinoids (TCs, capsaicin + dihydrocapsaicin) in the habanero pepper extract.

### 2.3.2. Extraction of Capsaicinoids Using Supercritical Fluids

Following a modified Shah et al. [24] methodology, the capsaicinoid extraction process began by weighing 40 g of habanero pepper powder. Ethanol was employed as a co-solvent with a concentration of 20% *w/w*. The powder, folded in a filter paper, was placed inside the extraction vessel (500 mL). Capsaicinoid extraction was carried out with supercritical fluid extraction equipment (SFT-150, Supercritical Fluid Technologies, Inc., Newark, DE, USA). The extractions were carried out using the static mode, whereby the equipment was adjusted to a predetermined pressure and temperature for a specific period of time according to the experimental design.

### 2.4. Determination of Capsaicinoids in Habanero Pepper Extract

The analysis of capsaicinoids was conducted using an ACQUITY UPLC H-Class System (Waters, Milford, MA, USA) equipped with a diode array detector (DAD). A column of ACQUITY UPLC HSS C18 (Waters, Milford, MA, USA) was employed for the analysis. A calibration curve (Figure S1) was made with Cp and DhCp (Sigma-Aldrich®, St. Louis, MO, USA). The mobile phase consisted of acetonitrile (mobile phase A) and water with 0.1% formic acid (mobile phase B) in a 60:40 ratio. The flow rate was maintained at 0.2 mL/min, and the column temperature was kept at 27 °C. For each analysis, an injection volume of 2 µL was used and the detection was performed at a wavelength of 280 nm. The TCs were quantified by adding the amounts of Cp and DhCp detected [23].

### 2.5. Determination of Antioxindat Capacity in Habanero Pepper Extract

To determine the antioxidant capacity (Ax) of extracts obtained through SFE, the DPPH method was employed, following the procedure outlined by Oney-Montalvo et al. [25]. In this method, 3.3 mg of DPPH was mixed with 100 mL of methanol. Then, the solution obtained was carefully adjusted to achieve an absorbance reading of $0.700 \pm 0.002$ at 515 nm using a Thermo Fisher Scientific®, Waltham, MA, USA, UV–Vis spectrophotometer (Genesys 140, México City, México). Subsequently, a 100 μL sample of the habanero pepper fruit extract was added to 3.9 mL of the adjusted DPPH solution. After mixing with vigorous agitation, the solution was left to incubate for 30 min. Following the incubation period, the absorbance (Abs) reading was measured at 515 nm. The antioxidant capacity (Ax) was then determined as the percentage of inhibition, calculated using Equation (1):

$$\% \text{ DPPH Inhibition} = 100 - [(\text{Habanero pepper extract Abs} \times 100)/(\text{Adjusted DPPH} - \text{solution Abs})] \qquad (1)$$

### 2.6. Statistical Analysis

The experiments were conducted using a randomized experimental $2^3$ factorial design. The data presented are expressed as means $\pm$ standard deviations, with capsaicinoids by triplicate and antioxidant capacity (Ax) by quadruplicate. The analysis was completed with a principal component analysis (PCA). Data analyses were performed using the statistical software Statgraphics Centurion XVII.II-X64 (Statgraphics Technologies Inc., Virgin, UT, USA), Excel (version 2108, Microsoft Corporation, Redmond, WA, USA), and R 4.0.3 (The R Foundation for Statistical Computing, Vienna, Austria).

## 3. Results

### 3.1. Capsaicinoids Content in Habanero Pepper Extract

To carry out the extraction of capsaicinoids from habanero peppers, a flour with a moisture content lower than 5% was used.

Once the extracts of habanero pepper obtained using supercritical fluids were analyzed, it was determined that the highest concentrations ($p < 0.05$) of Cp ($37.09 \pm 0.84$ mg/g Xt) and DhCp ($10.17 \pm 0.18$ mg/g Xt) were obtained under the conditions of 45 °C, 1450 psi, and an extraction time of 60 min. On the other hand, the lowest concentrations of Cp ($1.20 \pm 0.52$ mg/g Xt) and DhCp ($0.25 \pm 0.16$ mg/g Xt) were achieved at a temperature of 60 °C, a pressure of 2900 psi, and an extraction time of 60 min (Table 2).

**Table 2.** The $2^3$ factorial design for the evaluation of the capsaicinoid extraction conditions from habanero pepper using supercritical fluids.

| Exp | Factors | | | | | | Variable Response | | |
| | Coded Values | | | Real Values | | | Capsaicin (mg/g Xt) | Dihydrocapsaicin (mg/g Xt) | Total Capsaicinoids (mg/g Xt) |
| | $X_1$ | $X_2$ | $X_3$ | Tp (°C) | Ps (psi) | Et (min) | | | |
|---|---|---|---|---|---|---|---|---|---|
| 1 | −1 | −1 | −1 | 45 | 1450 | 60 | 37.09 ± 0.84 [h] | 10.17 ± 0.18 [g] | 47.26 ± 1.02 [h] |
| 2 | 1 | −1 | −1 | 60 | 1450 | 60 | 4.63 ± 0.13 [b] | 0.72 ± 0.02 [b] | 5.34 ± 0.16 [b] |
| 3 | −1 | 1 | −1 | 45 | 2900 | 60 | 15.62 ± 0.29 [d] | 2.48 ± 0.05 [c] | 18.09 ± 0.34 [d] |
| 4 | 1 | 1 | −1 | 60 | 2900 | 60 | 1.20 ± 0.52 [a] | 0.25 ± 0.16 [a] | 1.45 ± 0.67 [a] |
| 5 | −1 | −1 | 1 | 45 | 1450 | 120 | 14.43 ± 0.37 [c] | 2.30 ± 0.01 [c] | 16.73 ± 0.38 [c] |
| 6 | 1 | −1 | 1 | 60 | 1450 | 120 | 27.78 ± 0.04 [g] | 9.11 ± 0.02 [f] | 36.88 ± 0.06 [g] |
| 7 | −1 | 1 | 1 | 45 | 2900 | 120 | 24.00 ± 0.28 [f] | 7.61 ± 0.01 [e] | 31.61 ± 0.28 [f] |
| 8 | 1 | 1 | 1 | 60 | 2900 | 120 | 18.67 ± 0.1 [e] | 3.01 ± 0.05 [d] | 21.67 ± 0.06 [e] |

Note: Tp = Temperature; Ps = Pressure; Et = Extraction time; Xt = Extract. Different letters on each column show a statistically significant difference; Values are means $\pm$ SD ($n = 3$).

Consequently, the total content of capsaicinoids, taken as the sum of Cp and DhCp for each of the extracts obtained from the habanero peppers, shared the same extraction conditions in both the highest and lowest concentrations of these two metabolites.

The statistical analysis resulting from the analysis for each of the response variables (Cp, DhCp, TC) of the $2^3$ factorial experimental design were used to generate Pareto and interaction plots.

In Figure 1, it can be observed that all the main factors, as well as their double interactions (except Tp $\times$ Ps, $p > 0.05$) and their triple interaction showed a significant ($p < 0.05$) effect on the concentration of Cp found in the extracts.

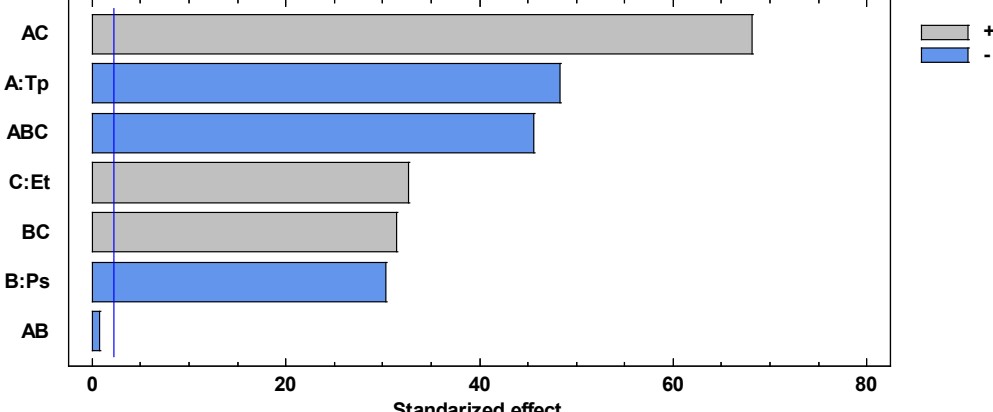

**Figure 1.** Pareto chart of capsaicin; Tp = Temperature; Ps = Pressure; Et = Extraction time.

In Figure 2, it can be observed that as the temperature increases (60 °C) and the extraction time decreases (60 min), the concentration of Cp in the habanero pepper extract decreases. Conversely, if the temperature decreases (45 °C) and the extraction time increases (120 min), the concentration of Cp increases. Regarding the interaction between pressure and extraction time, a strong negative effect can be observed on the concentration of Cp when the pressure increases from 1450 psi to 2900 psi and the extraction time decreases to 60 min.

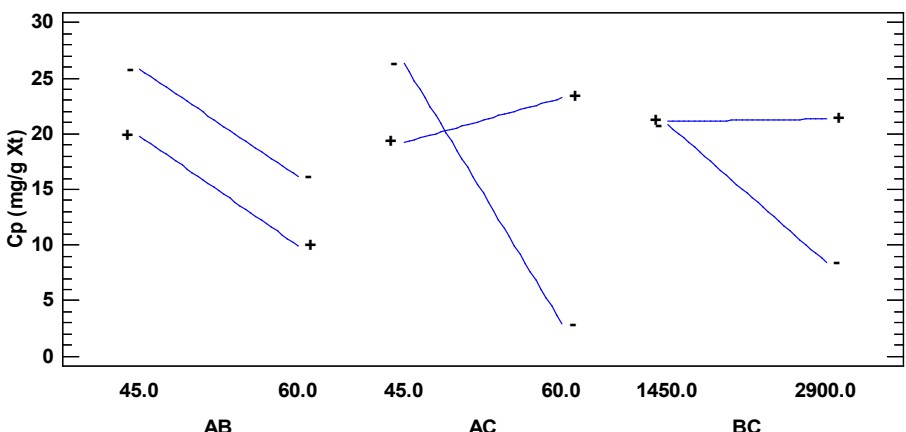

**Figure 2.** Interaction plot of extraction conditions for capsaicin (Cp); A = Temperature; B = Pressure; C = Extraction time; Xt = Habanero pepper extract.

The Pareto plot (Figure 3) and interaction plot (Figure 4) for DhCp were also obtained. In Figure 3, it can be observed that the triple interaction of the factors (Tp, Ps, and Et) had a significant effect on the concentration of DhCp in the habanero pepper extracts. Additionally, it is evident that both the main factors and their double interactions exhibit a significant effect ($p < 0.05$).

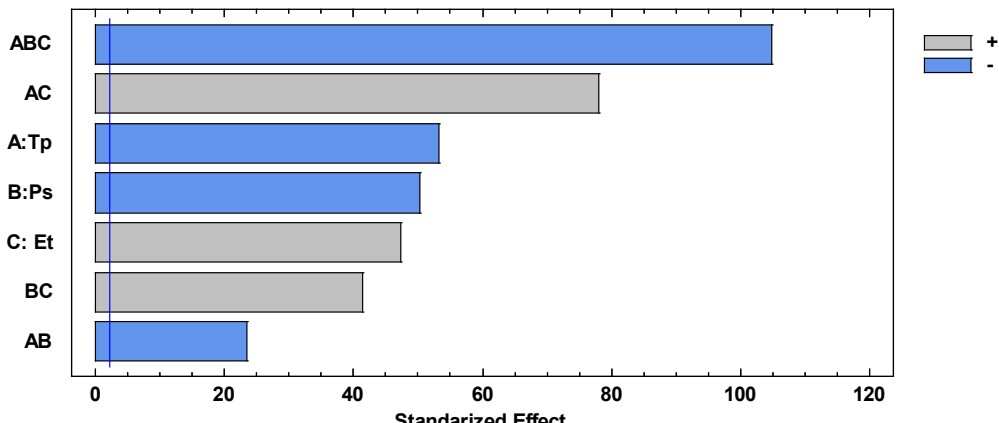

**Figure 3.** Pareto chart of dihydrocapsaicin; Tp = Temperature; Ps = Pressure; Et = Extraction time.

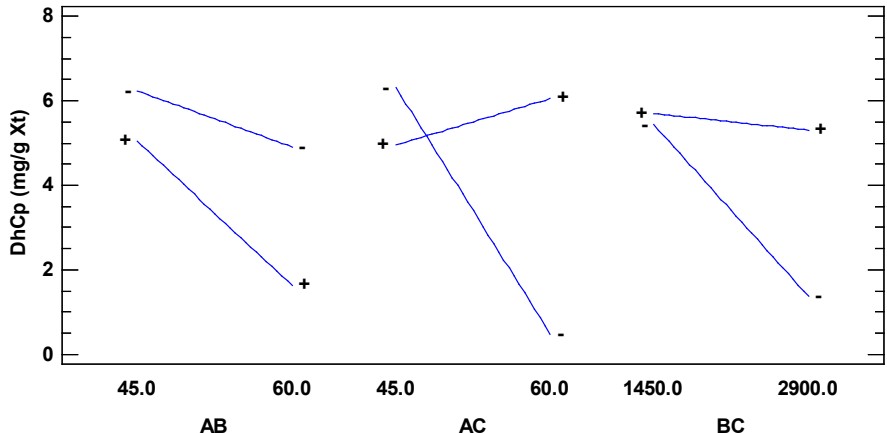

**Figure 4.** Interaction plot of extraction conditions for dihydrocapsaicin (DhCp); A = Temperature; B = Pressure; C = Extraction time; Xt = Habanero pepper extract.

By examining the interaction plot (Figure 4), it can be identified that the interaction between temperature and pressure tends to result in a lower concentration of DhCp whenever the temperature increases, regardless of the increase or decrease in pressure. The interaction between temperature and extraction time shows an increase in the concentration of DhCp in the extracts when Tp changes from 45 °C to 60 °C, and Et reaches 120 min, whereas the opposite behavior occurs when Et remains at 60 min. Lastly, the interaction between pressure and extraction time exhibits a similar effect on DhCp compared to Cp, as in both cases, the increasing pressure (from 1450 psi to 2900 psi) presented a negative effect on the concentration of these metabolites, regardless of Et.

The Pareto charts (Figure S2) and interaction (Figure S3) of TC presented similar data to those reported for Cp and DhCp.

### 3.2. Antioxidant Capacity from Habanero Pepper Extract

The highest ($p < 0.05$) antioxidant capacity (Ax) was observed in the extract obtained under the extraction conditions of 60 °C, 2900 psi, and 60 min, which exhibited the lowest concentration of all the metabolites evaluated in the present study (Cp, DhCp, TC). On the other hand, the lowest antioxidant capacity was obtained in the extract obtained under the conditions of 60 °C, 1450 psi, and 120 min (Table 3).

**Table 3.** The $2^3$ factorial design for the evaluation of extraction conditions' effects on antioxidant capacity from habanero pepper extracts obtained with supercritical fluids.

| Exp | Factors | | | | | | Variable Response |
|---|---|---|---|---|---|---|---|
| | Coded Values | | | Real Values | | | Antioxidant Capacity (% Inhibiton) |
| | $X_1$ | $X_2$ | $X_3$ | Tp (°C) | Ps (psi) | Et (min) | |
| 1 | −1 | −1 | −1 | 45 | 1450 | 60 | 87.73 ± 0.12 [g] |
| 2 | 1 | −1 | −1 | 60 | 1450 | 60 | 83.81 ± 0.18 [f] |
| 3 | −1 | 1 | −1 | 45 | 2900 | 60 | 53.07 ± 0.20 [c] |
| 4 | 1 | 1 | −1 | 60 | 2900 | 60 | 91.48 ± 0.24 [h] |
| 5 | −1 | −1 | 1 | 45 | 1450 | 120 | 50.96 ± 0.14 [b] |
| 6 | 1 | −1 | 1 | 60 | 1450 | 120 | 49.11 ± 0.29 [a] |
| 7 | −1 | 1 | 1 | 45 | 2900 | 120 | 69.48 ± 0.36 [e] |
| 8 | 1 | 1 | 1 | 60 | 2900 | 120 | 64.37 ± 0.07 [d] |

Note: Tp = Temperature; Ps = Pressure; Et = Extraction time; Xt = Extract. Different letters on each column show a statistically significant difference; Values are means ± SD ($n = 4$).

The statistical analysis of Ax showed (Figure 5) that all the main factors (Tp, Ps, Et), their double interactions, and their triple interaction presented an effect on the response variable (Ax). The main factors and their interactions that displayed an effect on the antioxidant capacity of habanero pepper extracts are Tp, Ps, Ps × Et, and Tp × Ps. On the other hand, Et, Tp × Et, and Tp × Ps × Et exhibited an effect on Ax.

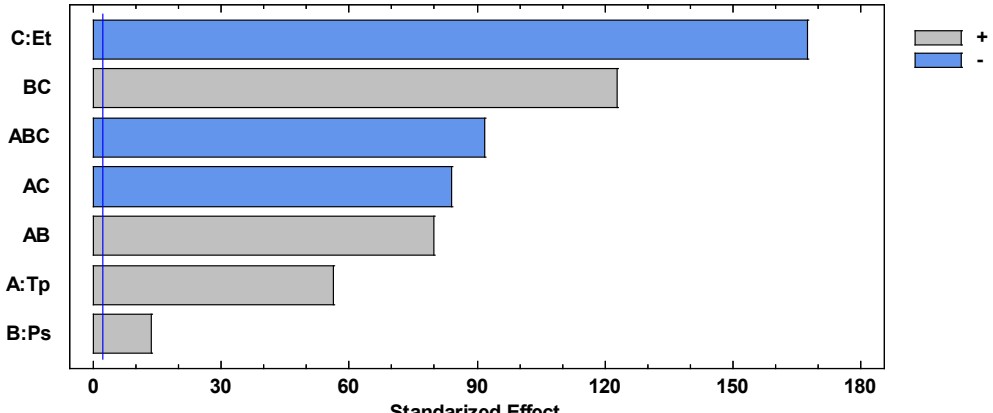

**Figure 5.** Pareto chart of antioxidant capacity (Ax); Tp = Temperature; Ps = Pressure; Et = Extraction time.

In Figure 6, the effect of the double interaction of the factors on Ax is observed. The antioxidant capacity of habanero pepper extracts increases when the temperature rises from 45 °C to 60 °C, at the pressure of 2900 psi. This behavior is also observed when the extraction temperature is 60 °C and the extraction process lasts 60 min. Finally, a high Ax can also be obtained when the pressure is maintained at 1450 psi, as long as the extraction process lasts 60 min.

### 3.3. Principal Component Analysis (PCA)

To complete the information obtained from the statistical analysis of the $2^3$ factorial design, a principal component analysis (PCA) was implemented, which allows exploration and identification of differences as well as similarities between the factors and response variables.

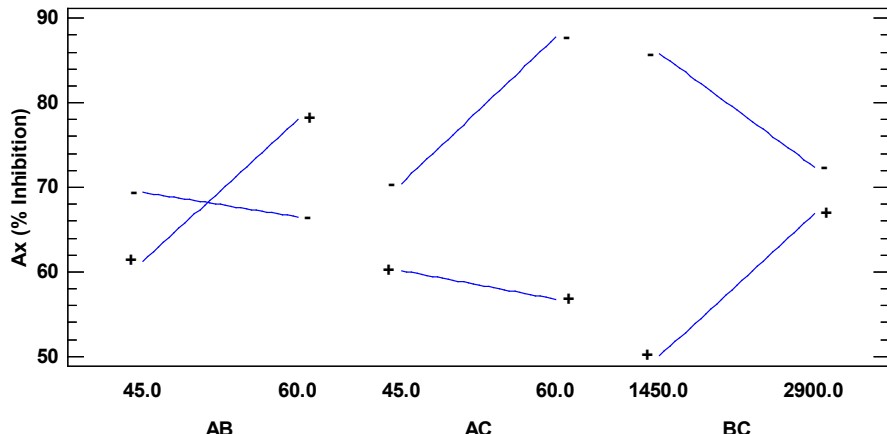

**Figure 6.** Interaction plot of extraction conditions for antioxidant capacity (Ax); A = Temperature; B = Pressure; C = Extraction time.

According to the PCA (Figure 7a), it can be observed that pressure and temperature exhibit an inverse behavior compared to the response variables. As Tp and Ps factors increase, concentrations of Cp, DhCp, and consequently TC decrease in habanero pepper extracts. It is also observed that pressure has a slight association with the evaluated metabolites in this study, but it does exhibit an inverse association with the antioxidant capacity of the extracts. Moreover, as the extraction time increases, the antioxidant capacity (Ax) of the extracts decreases, and vice versa. On the other hand, there is no association between Ax and Et with Cp, DhCp, and TC.

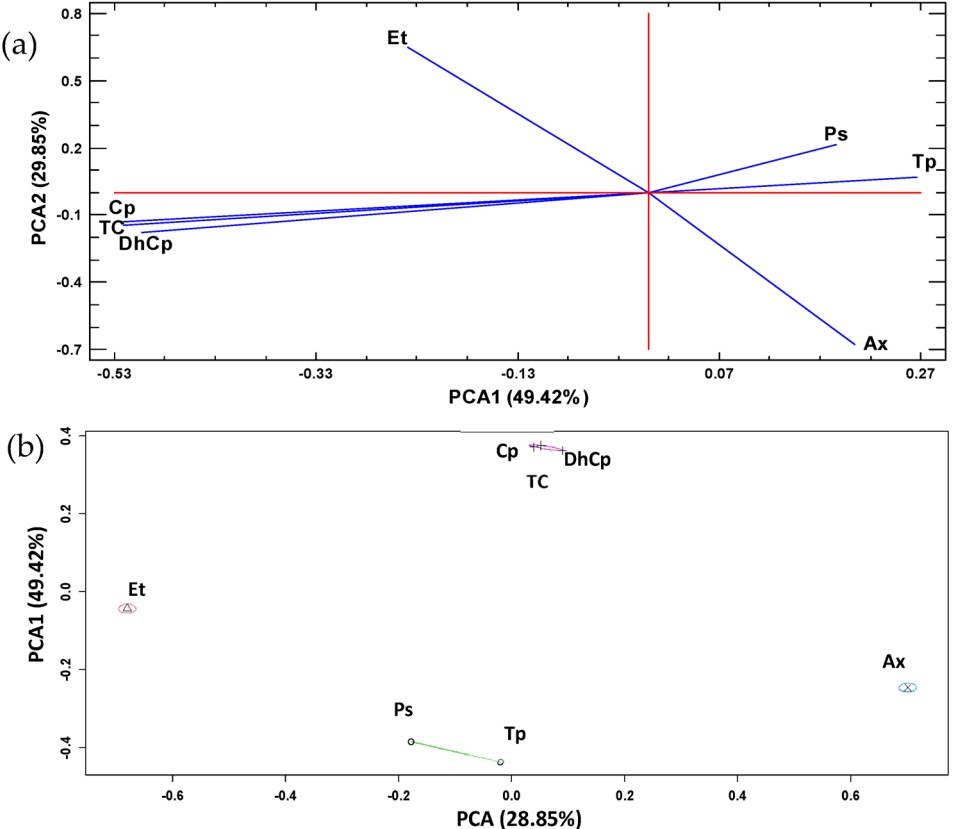

**Figure 7.** Analysis of capsaicinoids, antioxidant capacity, and extraction conditions; (**a**) Principal component analysis (PCA) and (**b**) Cluster of k-means; Abbreviations: Tp = Temperature; Ps = Pressure; Et = Extraction time; Ax = Antioxidant capacity; Cp = Capsaicin; DhCp = Dihydrocapsaicin; TC = Total capsaicinoid.

In Figure 7b, the cluster of k-means (number of clusters) is shown. The first cluster (green) consists of Tp and Ps, the second cluster (purple) includes Cp, DhCp, and TC, the third cluster (red) only includes Et, and the last cluster (blue) is composed solely of Ax. The above indicates that the metabolites exhibit a similar behavior when extracted using supercritical fluids, and both temperature and pressure are relevant factors during the extraction process, in contrast to the extraction time. It is also observed that Ax does not show any similarity with the evaluated metabolites' behavior in this study.

## 4. Discussion

According to the results of the $2^3$ experimental design, the triple interaction of the factors (temperature, pressure, and extraction time) showed a significant effect ($p < 0.05$) on CP, DhCp, TC, and Ax. By using a static mode in the SFE, and under the evaluated conditions, a temperature of 45 °C, pressure of 1450 psi, and an extraction time of 60 min exhibited the best results in terms of Cp concentration ($37.09 \pm 0.84$ mg/g Xt) and DhCp concentration ($10.17 \pm 0.18$ mg/g Xt) in the extracts of previously oven-dried habanero peppers. De Aguiar et al. [17] reported the extraction of Cp from malagueta peppers (*Capsicum frutescens*) using supercritical fluids, evaluating different temperatures (40, 50, or 60 °C) and pressures (2175.57, 3625.94, 5076.32 psi), and using 320 min (dynamic mode) from samples obtained through two different drying methods (freeze-drying, oven drying). The obtained concentration of Cp from the freeze-dried pepper sample ($42 \pm 2$ mg/g Xt) was similar to the reported value in this study. On the other hand, the oven-dried sample ($28.3 \pm 0.9$ mg/g Xt) showed a lower concentration. The concentration of DhCp has also been reported, showing a higher concentration for the freeze-dried sample ($19.9 \pm 0.3$ mg/g Xt) and a similar trend with the oven-dried sample ($14.4 \pm 0.3$ mg/g Xt) when comparing both data with those obtained in the present study ($10.17 \pm 0.18$ mg/g Xt). Both capsaicinoids were extracted under conditions of 40 °C, 2175.57 psi, and 320 min (dynamic mode). The difference observed in Cp concentration could be attributed, initially, to the thermal pre-treatment of the chili pepper. Another section of the study by Aguiar et al. [17] mentions that the chili was dried for 20 h at 70 °C, while in the present study, a temperature of 65 °C for 72 h was used. It has been reported that above 60 °C of drying temperature, the concentration of Cp decreases [26]. This suggests that longer drying times at a temperature slightly below 60 °C may contribute to a higher concentration of Cp. It is also important to consider the use of a co-solvent in the present study, specifically ethanol (20% ($w/w$)), that improves the extraction process due to the good solubility of Cp in this organic solvent [27,28]. However, the use of co-solvents can modify the characteristics of the solvent depending on the extraction conditions, such as in the case of $CO_2$. For instance, the density of the solvent has a directly proportional relationship with the pressure, meaning that the higher the pressure, the higher the density of the solvent. This, with the inherent characteristics of the co-solvent, leads to a rapid increase in particle size (turgor) resulting in a better penetration into the food matrix. This, in turn, facilitates a more efficient extraction of compounds that have higher solubility in the solvents [29].

Therefore, when using a co-solvent such as ethanol, it is not necessary to reach high pressures to obtain high concentrations of Cp, as demonstrated in the present study (1450 psi). The combination of $CO_2$ and the co-solvent enables a more effective extraction of Cp, even at low pressures, due to the enhanced solubility and penetration provided by the co-solvent [29].

In the present study, it was found that a suitable temperature for the extraction of capsaicinoids was 45 °C. This result is in agreement with several authors such as Santos et al. [15], de Aguiar et al. [20], Duarte et al. [19], Silva et al. [21], Rocha-Uribe et al. [22], and Yan et al. [30], who have also reported an extraction temperature around 40 °C for obtaining a high concentration of Cp. The effect of temperature on Cp extraction is minor compared to pressure. According to Knez et al. [31], the solvating capacity of $CO_2$ is primarily attributed to density, which is directly proportional to pressure and promotes an increase in the vapor pressure of the solute. However, when the pressure is slightly above the critical pressure (1058.77 psi), an



increase in temperature can result in a decrease in solvent density leading to a decrease in the mass transfer of the solute between the matrix and the solvent [32]. This explains the observed inverse relationship between pressure and capsaicinoid concentration in the present study.

It has also been reported that the increase in Et could promote a higher yield in the concentration of bioactive compounds extracted from the plant matrix. This is because the prolonged contact between the solvent and the sample allows the matrix to swell and the solvent to penetrate more easily, resulting in greater mass transfer. However, if the extraction time is excessively prolonged, it can lead to a degradation of the extracted bioactive compounds, resulting in low-quality extracts and leading to increased operating costs [33,34].

The highest Ax (91.48 ± 0.24% inhibition) in the habanero chili extracts was obtained under temperature conditions of 60 °C, pressure of 2900 psi, and an extraction time of 60 min. These results differed from those reported by Farahmandfar et al. [35], where the implemented conditions were a temperature of 35 °C, and 1450.38 psi with a 30 min extraction time for supercritical fluid extraction of phenolic compounds from *Capsicum frutescens*, resulting in a maximum DPPH inhibition of 56.56 ± 2.17%. However, these values are similar to those obtained (50.96 ± 0.14% inhibition) in the present study under conditions of 45 °C, 1450 psi, and 60 min of extraction, which yielded the lowest concentrations of Cp. Grande-Villanueva et al. [36], evaluated different temperatures (40 °C, 60 °C) and pressures (2175.75 psi, 2900 psi, 3625.94 psi, 4351.13 psi, 5076.32 psi) for the supercritical fluid extraction of jalapeño extract (*Capsicum annuum*). They observed that the extract with the highest concentration of total polyphenols (3.7 ± 3 mg GAE/g Xt) exhibited the highest antioxidant capacity (244 ± 21 μg GAE/g Xt) determined with the ferric reducing ability of plasma (FRAP) methodology. On the other hand, the extract with the lowest antioxidant capacity (176 ± 2 mg GAE/g Xt) had a low concentration (3.0 ± 0.1 mg GAE/g Xt) of total polyphenols. A similar behavior was observed in all the reported results and can be explained by the fact that the main bioactive compounds in habanero pepper, such as phenolic compounds, carotenoids, and vitamins, play a significant role in its antioxidant capacity in addition to capsaicinoids [25,35,37]. The extracts obtained in this study may vary in the antioxidant capacity due to the presence of diverse bioactive compounds, which could be simultaneously extracted along with capsaicinoids. This can explain why the antioxidant capacity (Ax) shows a low association (PCA) with Cp, DhCp, and TC.

Finally, it is possible to obtain extracts with high concentrations of Cp and a high antioxidant capacity by using ethanol as a co-solvent during the supercritical fluid extraction process. This is accomplished by reducing the pressure required to extract the capsaicinoids from habanero peppers, thus reducing operating costs and making supercritical fluid extraction of capsaicinoids more cost-effective. This, in turn, adds value to the habanero peppers from the Yucatán Peninsula [22].

## 5. Conclusions

The extraction of capsaicinoids from habanero peppers (*Capsicum chinense* Jacq.) through supercritical fluid extraction using ethanol (20% *w/w*) as a co-solvent demonstrated the possibility to obtain extracts with high concentrations of capsaicinoids and a high antioxidant capacity. This was achieved by utilizing extraction conditions below those reported in the literature, such as a pressure of 1450 psi and a temperature of 45 °C. It was also shown that both pressure and temperature are determining factors in the extraction of capsaicinoids during the supercritical fluid extraction process, while the extraction time directly affects the antioxidant capacity of the obtained extract. The reduction in temperature, pressure, and extraction time compared to reports in the literature for the extraction of capsaicinoids demonstrates that this process could be optimized under the conditions (45 °C, 1450 psi, 60 min) reported in this study. This optimization will result in reduced production costs and provide added value to the habanero peppers from the Yucatán Peninsula by obtaining high-purity capsaicin-rich extracts, which could be used as a bioactive ingredient for the development of pharmaceutical products or functional foods.

**Supplementary Materials:** The following supporting information can be downloaded at https://www.mdpi.com/article/10.3390/pr11082272/s1: Figure S1: Calibration Curve of Capsaicin from 0.005 to 0.08 mg/mL; $R^2$ = 0.9966; Figure S2: Pareto chart of total capsaicinoids; Tp = Temperature; Ps = Pressure; Et = Extraction time; Figure S3: Interaction plot of total capsaicinoids (TC); A = Temperature; B = Pressure; C = Extraction time; Xt = Habanero pepper extract.

**Author Contributions:** Conceptualization, I.M.R.-B. and K.A.A.-B.; methodology, K.A.A.-B. and I.M.R.-B.; software, K.A.A.-B. and I.M.R.-B.; validation, I.M.R.-B., M.O.R.-S., M.S. and G.F.; formal analysis, I.M.R.-B., M.O.R.-S. and M.S.; investigation, K.A.A.-B. and I.M.R.-B.; resources, I.M.R.-B.; data curation, I.M.R.-B. and M.S.; writing—original draft preparation, K.A.A.-B.; writing—review and editing, I.M.R.-B., M.O.R.-S. and M.S.; visualization, I.M.R.-B.; supervision, I.M.R.-B.; project administration, I.M.R.-B.; funding acquisition, I.M.R.-B. All authors have read and agreed to the published version of the manuscript.

**Funding:** This research was funded by the National Council of Science and Technology of Mexico (CONACYT), which financed the project No. 257588 and the scholarship 661099 for Kevin Alejandro Avilés-Betanzos.

**Data Availability Statement:** Not applicable.

**Conflicts of Interest:** The authors declare no conflict of interest.

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
