# Peer review of "Evaluation of the Capsaicinoid Extraction Conditions from Mexican Capsicum chinense Var. Mayapan with Supercritical Fluid Extraction (SFE)"

_processes, doi:10.3390/pr11082272_

Round 1
Reviewer 1 Report
-The main question is interesting and has relevance for the industrial production in México, but the topic is not original.
-The objective is clear, the information in the introduction is enough to understand the context.
-The experimental work is well designed and the statistical analyse is ok.
-The conclusions are consistent with the results obtained, only I want to do a comment about antioxidant activity:
In line 350 results and can be explained by the fact that the main bioactive compounds in Habanero pepper, such as phenolic compounds, carotenoids, and vitamins, play a significant role in its antioxidant capacity in addition to capsaicinoids [25,35,37].
356- This can explain why the antioxidant capacity (Ax) shows a low association (PCA) with Cp, DhCp, and TC.
Comment: With base in this allegation, it should have been done a composition analysis to identified which are the metabolites present in the extracts responsible by antioxidant activity.
-Tables and figures are well made.
Author Response
"Please see the attachment."

Reviewer 2 Report
Comments on: Processes (ISSN 2227-9717)
This manuscript focuses on optimizing the supercritical CO2 extraction of capsaicinoids from Mexican Capsicum chinense. The SFE technique is quite effective and environmentally sustainable for extracting secondary metabolites. I would like to get the author’s attention to the following comments:
The current title of the paper, "Evaluation of the SFE," is quite broad and does not clearly specify the nature of the evaluation conducted in the study. It would be beneficial for the authors to provide more clarity in the title by specifying the type of evaluation, such as experimental design or economic feasibility. Therefore, I encourage the authors to refine the title to accurately reflect the purpose and scope of their study.
Additionally, the introduction section could be further developed to include a more comprehensive comparative study. It would be beneficial to provide specific information about the operating conditions used in other capsaicin-rich extraction techniques. By including these details, readers will gain a better understanding of the comparative advantages and disadvantages of different extraction methods.
Experimental Design and Optimization:
The authors should provide an explanation of how they determined the three operating conditions (pressure, temperature, and time) used in their study. It would be helpful to clarify whether a screening design was conducted and if other factors, such as flow rate and co-solvent percentage/concentration, were considered. These additional parameters can significantly impact the efficiency and effectiveness of the extraction process and should be addressed.
Regarding the experimental design section, it is important to note that factorial design may not be the most appropriate method for studying the effect of operating conditions in the SFE process, especially considering the lack of information about the extraction of capsaicinoids using SFE and the uncertainties surrounding the curvature effect of the operating parameters. Alternative experimental design approaches may be more suitable and could be discussed.
Optimization of Co-solvent Percentage:
The authors should consider optimizing the co-solvent percentage as it is known to be a significant parameter in the SFE of secondary metabolites. By optimizing this parameter, the authors can enhance the economic feasibility of the process, which is an important aspect to consider in practical applications.
By addressing these points, the paper will benefit from improved clarity, enhanced comparability, and a more robust experimental design, leading to a more comprehensive evaluation of the SFE process for capsaicin-rich extractions.
The current paper demonstrates a satisfactory level of English proficiency.
Author Response
"Please see the attachment."

Reviewer 3 Report
This is a well-written article about an interesting and relevant topic that will contribute to the knowledge of supercritical CO2 extraction of capsaicionids, especially from the Habanero pepper originating from Yucatan area. However, minor revisions of the manucsript are needed for it to be suitable for publication. I enclose my comments further in text.
-Capsaicin was defined as Cp in line 15, so please use the abbreviation further throughout the abstract, also i would recommend adding the abbreviation for dihydrocapsaicin in the abstract
-Line 31 This sentence should be rephrased to be more clear and possibly merged with the conclusion of abstract.
e.g. The pressure and extraction time which yielded highest concentrations of cp and dihydrocapsaicin in the present study were lower than those reported in literature, indicating the possibility of reduced cost production and improved utilization of Capsicum chinense in the industry with further optimization processes.
-line 53, please define the capsaicin as Cp in this line since it is first mentioned here, and use the abbreviation further in text
-line 66 please replace the verb capable into efficient (capable is more often used when describing human qualities)
-line 93: What was the reason for using immature peppers (capsaicin concentration should be higher in ripe peppers?)? If there is a specific reason, please add a sentence to the introduction explaining it.
-line 112 please use a uniform name dyhydrocapsaicin or hydrocapsaicin throughout the text. This is the first time hydrocapsaicin appears so i would advise adding an abbreviation here which will be used further in text.
-line 135 please define the abbreviation of total capsaicinoids here and use it further in text
-line 144 change vigorously agitate to vigorous agitation
-Table 2 at exp, column dihydrocapsaicin you have an exponent 2, not a letter
-line 287 remove ''and'' before pressures, just add a comma
-lines 294-297 please check if the conditions and values you mention are in accordance to the reference
-lines 304-310 please add reference
lines 353-355 rephrase the sentence, it is rather unclear
-lines 374-377 What do you mean by optimized at these conditions? Do you mean other parameters relevant for the extraction process, while these are kept constant? If so, please make more clear.
-line 384 remove the MDPI sentence and leave the authors contribution only. Same for lines 391-393
-throughout the manuscript, make sure that all abbreviations are defined when the term is mentioned for the first time and use them further on instead of full word
English language is mostly fine, some sentences (mentioned in my comments) should be rephrased.
Author Response
"Please see the attachment."

Reviewer 4 Report
The comments are as follows:
1. The authors should provide moisture content of dried sample material.
2. At which temperature capsaicinoids start to degrade?
3. The authors are encouraged to express more the practical applications of the findings in the Conclusions section.
Author Response
"Please see the attachment."

Round 2
Reviewer 2 Report
Thanks for addressing the comments.